# Physiological, Biochemical, and Genetic Reactions of Winter Wheat to Drought Under the Influence of Plant Growth Promoting Microorganisms and Calcium

**DOI:** 10.3390/microorganisms13051042

**Published:** 2025-04-30

**Authors:** Mariam Zareyan, Rima Mockevičiūtė, Sigita Jurkonienė, Virgilija Gavelienė, Algimantas Paškevičius, Vaidevutis Šveikauskas

**Affiliations:** Laboratory of Plant Physiology and Laboratory of Biodeterioration Research, Nature Research Centre, Akademijos Str. 2, 08412 Vilnius, Lithuania; rima.mockeviciute@gamtc.lt (R.M.); virgilija.gaveliene@gamtc.lt (V.G.); algimantas.paskevicius@gamtc.lt (A.P.); vaidevutis.sveikauskas@gamtc.lt (V.Š.)

**Keywords:** *Bacillus subtilis*, calcium salts, drought stress, plant growth-promoting microorganisms (PGPMs), *Triticum aestivum*

## Abstract

Improving wheat drought stress tolerance is a critical and challenging task, and more research is necessary since many parts of the world depend on this crop for food and feed. Our current work is focused on the influence of probiotic microorganisms in combination with calcium salts on the physiological and biochemical metabolic pathways that wheat uses when exposed to drought stress and on the analysis of gene expression levels that contribute to wheat drought tolerance. The research was conducted in the laboratory under controlled conditions, simulating a prolonged drought. Seedlings were treated with different microorganisms (*Bacillus subtilis*, *Lactobacillus paracasei*, and some yeast) in 10^5^ CFU/mL concentrations for seed priming and later in the same concentration for seedling spraying. A total of 70 g/m^2^ CaCO_3_ or 100 g/m^2^ CaCl_2_ was added to the soil before sowing the seeds. Almost all tested treatments improved plant growth and positively affected prolonged drought resistance in winter wheat. *Bacillus subtilis*, in combination with calcium salts, had the greatest effect on maintaining the relative leaf water content (RWC). The proline, malondialdehyde (MDA), and H_2_O_2_ tests proved the significant positive impact of the treatments on the plant’s response at the biochemical level, with growth parameters close to those of irrigated plants, for example, the ones treated with *B. subtilis* alone or with Ca salts had the lowest H_2_O_2_ content, 0.86–0.96 μmol g^−1^ FW, compared to 3.85 μmol g^−1^ FW for the Control, along with lower levels of drought-induced gene expression. All the presented results show statistically significant differences (*p* < 0.05). This study showed that tested microorganisms in combination with calcium salts can activate plants’ defense reactions in response to drought. The practical significance of this study is that these ecological measures can be useful under field conditions.

## 1. Introduction

Drought is the most critical abiotic limiting factor for plant cultivation in many parts of the world [1,2,3,4,5]. Drought affects plants’ morphological, physiological, biochemical, and molecular processes, resulting in growth inhibition [6,7,8]. A bioinformatic analysis of European crop losses over recent decades revealed that drought waves are responsible for greater yield loss in cereals (9% and 7.3%) compared to non-cereal crops (3.8% and 3.1%) [9]. Many studies have explored the genetic and molecular mechanisms behind drought resistance in cereals such as wheat, barley, maize, sorghum, and rice [10,11]; however, as a result, comparably few solutions were offered that proved to be effective for organic farming.

Wheat is the leading food for over a third of the world’s population [12]. Wheat production in recent years (2018–2025) has been lower than it used to be in the largest producing countries such as Canada, Iran, France, Germany, and Greece due to drought [13,14,15]. According to the European Drought Observatory (EDO), a drought warning was in effect for nearly half of the EU territory in 2022, with a red alert for 15%. By mid-January 2025, the Combined Drought Indicator (CDI) showed warning drought conditions in southern Italy, the eastern Baltic Sea region, eastern Poland, Belarus, central-eastern Ukraine, Greece, part of the Balkans, Cyprus, Malta, and other Mediterranean islands, as well as Ireland, northern UK, and more than a half of Türkiye [16]. Therefore, developing new pathways to help plants overcome drought stress is essential.

Nutrient management and organic amendments can reduce the harmful effects of drought stress on crops [17]. As an essential plant nutrient, calcium plays a vital role in plant growth and development. Drought stress is known to alter plant mineral status and metabolic processes. Calcium (here and later Ca) is involved in plant signaling responses to drought [18] and thereby positively affects plant adaptation to drought stress [19]. Ca^2+^ can help plants adapt to drought quickly by adjusting their stomatal opening/closing, optimizing their gas exchange, and improving their photosynthetic efficiency [20]. Probiotic microorganisms increase plants’ uptake of essential elements from the soil. For example, microbial biostimulants enhance plant growth under stress conditions by facilitating the uptake of insoluble elements and essential minerals [21]. Probiotic bacteria are also thought to induce drought tolerance in plants by altering root morphology, activating the antioxidant system, and promoting the expression of genes associated with abiotic stress tolerance [22]. Plant growth-promoting microorganisms (PGPMs) use different mechanisms that directly or indirectly help in crop plants’ growth and development, increasing crop productivity [23]. *Bacillus subtilis* is one of the most attractive probiotics for developing natural plant protection products because it is generally considered safe for food use [24]. *B. subtilis* strains produce cytokinin, which interferes with the drought-induced suppression of shoot growth, thereby increasing plant growth during periods of drought [25].

Several drought tolerance indicators have been identified as reliable drought resilience measures, including relative water content (RWC), water use efficiency, stomatal conductance, photosynthesis, and drought-responsive genes. The RWC is considered a measure of plants’ water status and the dehydration index. It allows one to assess plants’ physiological states and shows their ability to retain water in unfavorable conditions [26,27]. Malondialdehyde (MDA) is commonly used to assess the status of redox and osmotic adaptation, which is important for plant adaptation to environmental stresses. MDA is a good marker of oxidative stress because ROS breaks down polyunsaturated lipids and forms MDA. The overall increase in membrane lipid peroxidation is proportional to the intensity of drought stress and may result from spontaneous reactions of ROS with organic molecules contained in membranes [28]. Among ROS species, hydrogen peroxide (H_2_O_2_) is an especially important one involved in plant responses to different environmental stressors. During stress, peroxide concentration increases because it works as a signaling molecule. However, increasing H_2_O_2_ concentration over time can harm plants (oxidative stress) [29]. In stressful conditions, plants accumulate metabolites, especially some amino acids such as proline. Proline plays the role of the osmolyte for plants; during drought stress, it can take the role of a signaling molecule, a metal chelator, and an antioxidant defense molecule. It is often used as a stress marker [30,31]. Plants respond to stressful environments by altering gene expression and protein production [6]. It is known that some genes that code late embryonic protein (LEA) are affected by drought [32,33]. LEA proteins are generally predicted to maintain intrinsically disordered proteins in a fully hydrated state, which can then fold under water-deficient conditions to adopt α-helical structures. Ordered LEAs can act as molecular chaperones to bind enzymes, membranes, water, ions, and reactive oxygen species (ROS) and, therefore, play a critical role in protein/membrane stabilization and maintaining cellular environmental homeostasis under drought stress [33].

The experiments were modeled and conducted with the microorganisms *Bacillus subtilis*, *Lactobacillus paracasei*, *Zygosaccharomyces bailii*, and *Geotrichum silvicola*, some of which have never been investigated separately for their potential as plant growth-promoting microorganisms, or in the case of *B. subtilis,* it has never been studied in combination with Ca in drought stress. This study aimed to investigate how the effects of different microorganisms and Ca could modify the drought tolerance of wheat at the molecular, biochemical, and morphological levels. Further work focused on determining which of the selected microorganisms help the plant overcome drought stress, especially in the presence of added Ca in the soil. The objectives of this study were to (1) explore whether chosen microorganisms and Ca salts improve shoot growth under drought stress, (2) study the changes in biochemical and molecular responses of treated winter wheat to drought, and (3) investigate which microorganisms work better in combination with Ca.

## 2. Materials and Methods

### 2.1. Plant Material, Growth Conditions, and Treatments

Wheat seeds (*Triticum aestivum* L. cv. ‘Skagen’) were sown in plastic cylindrical pots (10 × 13 cm), 20 seeds per pot, in a peat moss substrate SF1 from SuliFlor (pH 5.5–6.5). Plants were germinated and grown under controlled conditions of a constant temperature of 22 ± 1 °C, a photoperiod of 16/8 h, and a fluorescent light photon flux of 60 μmol m^−2^ s^−1^ at the soil level. Soil moisture was maintained at 60–70%.

The following treatments were used for the drought stress control studies:

(a) Ca was added to the soil in the form of CaCO_3_ at a rate of 1.3 g per pot, based on 70 g m^−2^, and in the form of CaCl_2_ prediluted in water at a rate of 1 g in 100 mL per pot.

(b) The microorganisms *Bacillus subtilis*, *Lactobacillus paracasei*, *Zygosaccharomyces bailii*, and *Geotrichum silvicola* were used for seed priming in 10^5^ CFU/mL concentrations (microorganisms were grown in their specific liquid media (*Bacilus subtilis* in Nutrient media, *Lactobacillus paracasei* in MRS media, and the yeast in YPD media to the log phase and then diluted with distilled water to obtain the required concentration), and later, in the same concentration for seedlings spraying in the 1–2 leaf stage (BBCH-scale 1–2) [34].

### 2.2. Experimental Design and Drought Conditions

Treatments were added according to the scheme (Table 1). Each treatment was carried out in 6 pots, divided into two groups: 3 for simulating drought and 3 for rational watering. Drought was initiated immediately after plants were sprayed with microorganisms at the 1–2 leaf stage as irrigation termination. As a result, the soil gradually dried, reaching 15–20% humidity. Soil humidity of watered plants was kept at a 60–70% level. Soil moisture was measured using a soil moisture meter (Aicevoos, AS-PH3, Shanghai, China).

### 2.3. Sampling

Plant samples were collected for the analysis when the soil moisture in the drought treatments was 15–20%. The watered plants were sampled simultaneously (soil moisture 60–70%). Shoots of wheat seedlings were sampled for morphometrical measurements: 10 plants from each pot. For biochemical and molecular analysis, three independent replicates were carried out using the leaves of wheat plants. For MDA, H_2_O_2_, proline assays, and RNA isolation, the samples were collected using liquid nitrogen and stored in a low-temperature freezer (Skadi Green line, International Labmate Ltd., St Albans, UK) at −80 °C until the analysis.

### 2.4. Morphometrical Measurements

Shoot length and shoot biomass were measured using a ruler and balances (Kern EWJ, Balingen, Germany).

### 2.5. RWC

RWC was determined according to Weng et al. [35]. Fresh wheat leaves were collected and weighed as fresh weight (FW). After that, the leaves were left in the water for 24 h and weighed again to obtain a saturated weight (SW). The dry weight (DW) was obtained by drying the leaves in a drying chamber and weighing. RWC was calculated according to the following formula [35]:RWC = [(FW − DW)/(SW − DW)] × 100%,(1)

### 2.6. Assessment of Biochemical Parameters

#### 2.6.1. H_2_O_2_

For analysis of H_2_O_2_ and MDA, leaf material (0.5 g) was homogenized using 5% trichloracetic acid (TCA) (Sigma-Aldrich, St. Louis, MO, USA). The homogenates were centrifuged at 10,000× *g* for 15 min (centrifuge MPW-351 R). H_2_O_2_ content in leaves was determined according to Velikova et al. [36]. The supernatant was mixed with 10 mM, pH 7.0 potassium phosphate buffer (Alfa Aesar), and 1 M potassium iodide (Alfa Aesar) in a ratio of 1:1:2. The reaction solution was incubated for 30 min at 25 °C in the dark. The absorbance of the supernatant was measured at 390 nm by a spectrophotometer (Analytik Jena Specord 210 Plus, Analytik Jena, Jena, Germany). The amount of H_2_O_2_ was calculated using a standard curve. The results are expressed in μmol g^−1^ FW.

#### 2.6.2. Lipid Peroxidation According to MDA

The method of Hodges et al. [37], with slight modifications, was used to estimate MDA. The supernatant was added to 20% TCA containing 0.5% thiobarbituric acid (TBA) (Alfa Aesar, Haverhill, MA, USA). The homogenate was incubated in a heater (Blockthermostat BT 200) at 95 °C for 30 min and then cooled on ice. The optical density was measured at 532 and 660 nm by a spectrophotometer (Analytik Jena Specord 210 Plus, Analytik Jena, Jena, Germany). The results are expressed in μmol g^−1^ FW [37].

#### 2.6.3. Proline

Carillo & Gibon’s proline extraction and determination method was used to determine the free proline content [38]. Leaf material (0.5 g) was homogenized using 40% ethanol and left overnight at +4 °C. The homogenates were centrifuged at 14,000× *g* for 5 min (centrifuge MPW-351 R), and the supernatant was added to the reaction mix (1% ninhydrin, 60% acetic acid, 20% ethanol). The homogenate was incubated in a heater (Blockthermostat BT 200) at 95 °C for 20 min. The optical density was measured at 520 nm by a spectrophotometer (Analytik Jena Specord 210 Plus, Analytik Jena, Jena, Germany). The corresponding content of the proline was determined using the standard curve. The calculations were made using the equation given in the method of [38]. The results were expressed in μmol g^−1^ FW.

### 2.7. Molecular Techniques

#### 2.7.1. RNA Extraction and Reverse Transcription

Total RNA was extracted from 200 mg of plant leaf material using the PureLink RNA Mini Kit (Ambion, Waltham, MA, USA) and Heraeus Fresco 21 Centrifuge (Thermo Scientific, Waltham, MA, USA). To avoid contamination with genomic DNA, extracted total RNA was treated with the RapidOut DNA Removal Kit (Thermo Scientific). The concentration and purity of treated RNA were evaluated with the spectrophotometer NanoPhotometer P330 (IMPLEN, Westlake Village, CA, USA). DNase-treated RNA samples were reverse-transcribed using the High-Capacity cDNA Reverse Transcription Kit (Applied Biosystems, Waltham, MA, USA) following the manufacturer’s recommendations. The obtained cDNA was stored at −20 °C.

#### 2.7.2. Real-Time Quantitative PCR

Real-time quantitative PCR was carried out using SYBR^®^ Green Universal Master Mix kit (Applied Biosystems), as recommended by the producer company, and by employing QuantStudioTM 5 real-time PCR system (Applied Biosystems). Two microliters of cDNA (the equivalent of 25 ng of total RNA) were used as a template for PCR. Cycling conditions comprised 1 cycle at 95 °C for 2 min and 40 cycles at 95 °C for 15 s, followed by 60 °C for 1 min. After the PCR run, a melting curve was generated and analyzed each time with the Quant Studio Design & Analysis Software v.1.5.2 (Applied Biosystems). Gene expression was calculated using the 2^−ΔΔCt^ method [39].

#### 2.7.3. Primers

The sequences of primers used in the work were based on the sequences of LEA protein genes and were taken from the publication of Ali-Benali et al. [40]. The wheat mitochondrial *26S* ribosomal RNA gene was used as a housekeeping gene [40]. The primer concentrations in all cases were 200 nM except for the *Td27e*, where the concentration was 900 nM. The sequences of all primers used in the work are listed in Table 2.

### 2.8. Statistical Analysis

The results are presented as mean ± standard deviation (SD) of three independent experiments with at least three replicates. The data were analyzed using the Microsoft Office Excel 2019 software, and a two-way analysis of variance with replication (two-way ANOVA with replication) was used. The significant differences between treatment means were determined using analysis of variance and mean separation at a 5% significance level (*p* ˂ 0.05).

## 3. Results

### 3.1. Impact of PGPMs and Ca Salts on Morphometric Parameters of Wheat Exposed to Prolonged Drought Stress

The shoot length and biomass of wheat were significantly higher when the plants were treated with probiotic microorganisms and grown in soil supplemented with Ca salts, especially in those with the combination of *B. subtilis* and CaCO_3_. This was seen in both drought-stressed and normally watered plants (Table 3).

### 3.2. Effect of Used PGPMs and Ca Salts on RWC of Wheat Seedlings Exposed to Prolonged Drought

Wheat plants treated with PGPMs and Ca salts retained higher water content in their leaves after exposure to prolonged drought (Figure 1). Those with a combination of *B. subtilis* and Ca salts had the best retention of leaf water content at ~80% (moderate stress). In comparison, the RWC of Control wheat leaves was 47% (high-stress level).

### 3.3. Effect of PGPMs and Ca Salts on Biochemical Responses of Wheat Plants Exposed to Prolonged Drought

#### 3.3.1. Hydrogen Peroxide (H_2_O_2_)

The levels of H_2_O_2_ increased in drought-affected plants (Figure 2). However, H_2_O_2_ content was found to be significantly lower in all variants treated with PGPMs and Ca salts in combinations. Those treated with *B. subtilis* alone or with Ca salts had the lowest H_2_O_2_ content, 0.86–0.96 μmol g^−1^ FW, compared to 3.85 μmol g^−1^ FW for the Control. The difference is also significant when comparing plants only treated with *Bacillus subtilis*, *Lactobacillus paracasei*, or Ca salts.

#### 3.3.2. Malondialdehyde (MDA)

MDA levels that indicate the lipid peroxidation levels showed that although drought stress significantly increased MDA content in wheat leaves, the amount of MDA was considerably lower in treated plants, especially in plants with a combination of *B. subtilis* and Ca salts (18–22 μmol g^−1^ FW) compared to the Control plants (72.4 μmol g^−1^ FW). The difference is also significant when comparing plants only treated with *B. subtilis* (31.013 μmol/g^−1^ FW) or with Ca salts (47–49.4 μmol g^−1^ FW). Combinations of *Lactobacillus paracasei* and CaCl_2_ (28.47 μmol g^−1^ FW) and *Zygosaccharomyces bailii* and CaCl_2_ (21.96 μmol g^−1^ FW) were also very effective in reducing MDA levels (Figure 3).

#### 3.3.3. Free Proline

The amount of free endogenous proline in the Control plants exposed to drought stress was 8.88 μmol g^−1^ FW. In contrast, in plants treated with PGPMs and Ca salts, it ranged from 1.97 to 6.1 μmol g^−1^ FW (except *Lactobacillus paracasei* + CaCO_3_, which was closer to the Control plants), the lowest in the samples treated with the combinations of *Bacillus subtilis* and CaCl_2_ and *Zygoaccharomyces bailii* and CaCl_2_ (Figure 4).

### 3.4. Effect of PGPMs and Ca Salts Application on Late Embryogenesis Abundant (Lea) Genes Expression Levels in Wheat Plants Exposed to Prolonged Drought

The quantitative results of the RT-qPCR analysis showed that among the three studied genes, the *Td27e* gene was more highly expressed in drought stress. The level of *Td27e* expression in the Control sample was significantly higher than in all treated types (Figure 5a). *Td29b* gene expression levels were markedly lower in all treated variants compared with the Control (Figure 5b). The lowest levels of expression of the *Td11* gene were recorded in the samples of plants treated with the combinations of *Bacillus subtilis* and Ca salts and *Geotrichum silvicola* and Ca salts (Figure 5c).

## 4. Discussion

According to the Population Division of the United Nations, the world human population will reach about 9.5 billion by 2050. Agriculture is the primary source of food production, and it must solve this problem by producing enough food for the global population [41].

This work is a logical continuation of the previous work [42]. Since a positive reaction of plants under drought conditions to the combination of commercial plant probiotics and CaCO_3_ was shown, it was important to understand which PGPMs were at work here and what role Ca played. Therefore, experiments were modeled and conducted with the following ingredient microorganisms (or related to them): *Bacillus subtilis*, *Lactobacillus paracasei*, *Zygosaccharomyces bailii*, and *Geotrichum silvicola*. In addition to CaCO_3_, CaCl_2_ was added to the experimental model as a more easily absorbed form of Ca to compare which works better in combination with microorganisms. A series of experiments were performed to explore how the application of probiotic microorganisms and Ca affects the response of wheat to drought stress at the physiological, biochemical, and molecular levels. A test experiment was designed and performed to exclude the possibility that minerals, sugars, or other components of microbial media have a more significant positive effect on plants than microorganisms. The results showed that microorganisms are essential for obtaining the previously observed positive impact. Further work focused on determining which of the selected microorganisms help the plant overcome drought stress, especially in the presence of added Ca in the soil.

Much research has been carried out to understand how plants respond to drought and how to optimize that response [28,29,31,43]. Thanks to previous research, we were able to select some of the most significant markers that show the level of stress in plants. For example, the RWC results showed that our treatments stimulated plants’ ability to retain water and subsequently maintain their physiological state close to that of the watered Control plants (Figure 1). The proline, MDA, and H_2_O_2_ tests (Figure 2, Figure 3 and Figure 4), in turn, proved that the treatments had a significant positive impact on the plant’s response at the biochemical level. Some of the PGPMs used with Ca salts maintained the concentrations of these stress indicators at almost half those of untreated controls. The MDA concentration difference is significant when comparing plants only treated with *B. subtilis* or in combination with Ca salts (Figure 3), which indicates that the combination of them is the most beneficial for the plant. The same is visible with the H_2_O_2_ test results (Figure 2) when comparing plants only treated with *Bacillus subtilis, Lactobacillus paracasei*, or in combination with Ca salts, and with the combinations of *Bacillus subtilis* and CaCl_2_ and *Zygoaccharomyces bailii* and CaCl_2_ (Figure 4) for the proline test. The poline test showed that the concentration of endogenous proline was four times higher in the Control plants compared to those treated with the combination of *B.subtilis* and CaCl_2_ (Figure 4); less proline, in this case, indicates reduced stress. The three genes that we used for our studies have a major role in wheat’s response to drought stress in the early stage of growth. *Td27e* and *Td11* code for group 2 LEA proteins, also named dehydrins. Dehydrins are produced by plants in response to cold and drought stress and are well-known as stress proteins [40,44]. The *Td27e* gene encodes YSK2-type dehydrin, and the *Td11* gene codes SK3-type dehydrin. The *Td29b* gene codes for a group 4 LEA protein and plays an important role in the early stages of seed development by protecting cells against damage caused by desiccation [40]. In this study, these genes were used as stress indicators to see the level of changes at the molecular level in plants and the effects of treatments at that level. As a result, their expression levels proved to be significantly lower when plants were treated with chosen PGPMs and Ca salts (Figure 5a–c).

Several studies have been conducted on using plant growth-promoting microorganisms under abiotic stress, especially from the *Bacillus* genus, and they have shown great potential for organic farming [5,45,46]. Previous research has already shown that *B. subtilis* plays an important role in wheat’s response to adverse environmental factors [47]. Based on our experiments, we can confirm that *B. subtilis* helped winter wheat plants maintain growth even under prolonged drought conditions and maintained their biochemical and molecular homeostasis close to that of plants watered rationally. Moreover, we see that in combination with Ca salts, the positive effect of *B. subtilis* increases significantly.

In contrast to the *Bacillus* genus, several studies have been conducted in animal models that have linked the genus *Lactobacillus* and the efficiency of Ca absorption in their presence [48,49,50]. An interesting piece of research connected the use of Ca supplements with *L. paracasei* colonization in the gut [51]. These studies and our results allow us to hypothesize that using *Lactobacillus* species in combination with Ca may also positively affect plants, but further research is necessary to clarify the essence of this combination.

Some other studies have linked yeast (*Zygosaccharomyces* genus) use to improve yield growth under abiotic stress conditions [52,53]. Our results showed great potential for using *Zygosaccharomyces bailli* in combination with Ca salts to alleviate the impact of drought on winter wheat plants. *Geotrichum silvicola*, which is not a well-investigated yeast, also showed this potential.

The results of our research allow us to propose several hypotheses regarding the connection of the use of PGPMs in combination with Ca for further investigation of possible pathways related to their effect on plants during drought. The current winter wheat study data showed that using these treatments can activate the defense responses of plants, which can compensate for the negative impact of drought. This, again, proves the results of previous research [41,54] that plant growth-promoting microorganisms have great potential to be used in agriculture.

## 5. Conclusions

The exogenous application of our PGPMs and Ca salts improved the prolonged drought tolerance of winter wheat. *B. subtilis*, in combination with Ca salts, had the most significant impact on maintaining the relative water content of the leaves and keeping plant growth parameters close to those of irrigated plants. This treatment also showed the best results for maintaining plants’ biochemical and molecular homeostasis. The results of this study for winter wheat showed that using these PGPMs in combination with Ca salts can activate defense reactions that can compensate for the harmful effects of drought stress and, therefore, stimulate their growth under drought conditions. The PGPMs and Ca salts used in this study are suitable for organic farming since they do not harm the environment; therefore, further research is necessary to evaluate their effectiveness under field conditions and on other plant species.

## Figures and Tables

**Figure 1 microorganisms-13-01042-f001:**
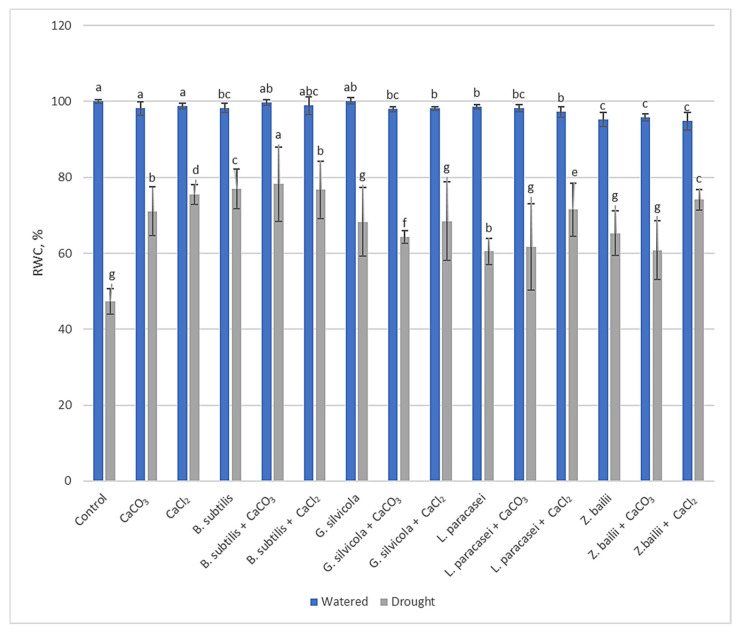
Impact of PGPMs and Ca salts on RWC of winter wheat leaves after drought stress. Error bars represent the standard deviation of the mean. Means with different letters in the same color columns are significantly different (*p* < 0.05).

**Figure 2 microorganisms-13-01042-f002:**
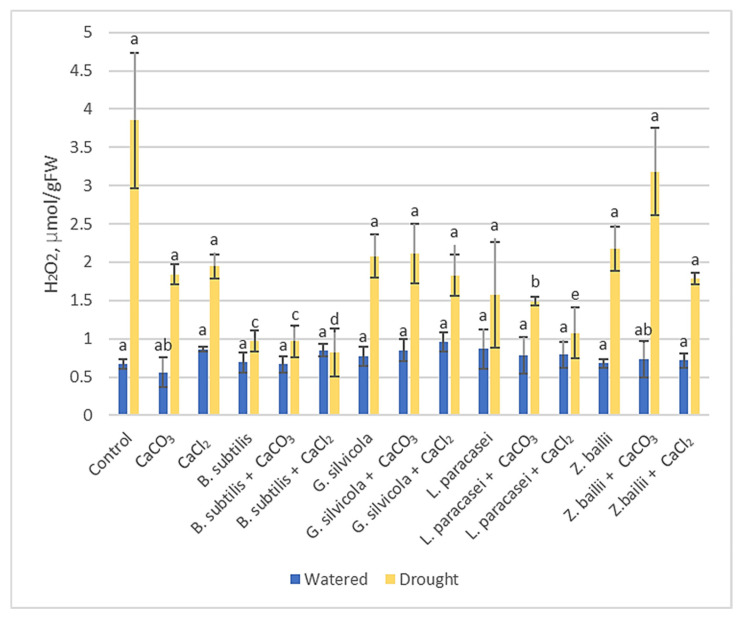
The effect of PGPMs and Ca salts application and prolonged drought stress on winter wheat H_2_O_2_ content. Error bars represent the standard deviation of the mean. Different lowercase letters in the same color columns indicate statistically significant differences (*p* < 0.05).

**Figure 3 microorganisms-13-01042-f003:**
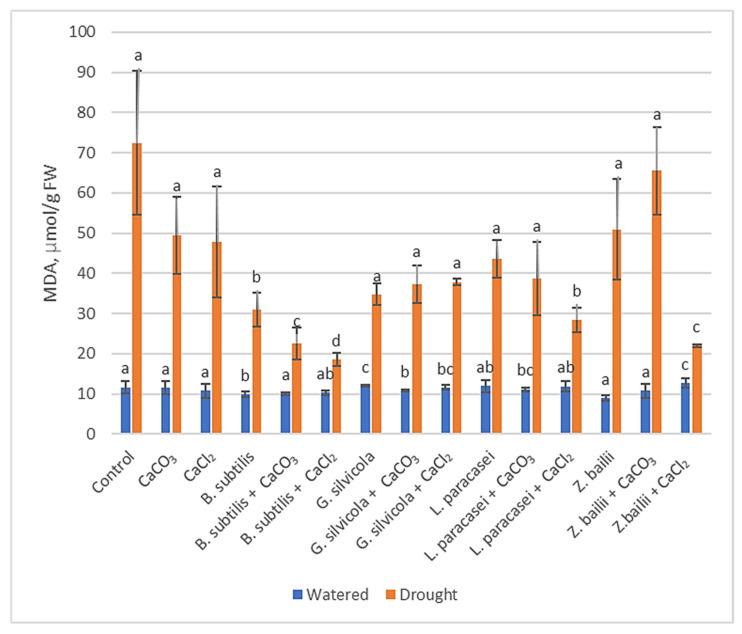
The effect of PGPMs and Ca salts application and prolonged drought stress on winter wheat MDA content. Error bars represent the standard deviation of the mean. Different lowercase letters in the same color columns indicate statistically significant differences (*p* < 0.05).

**Figure 4 microorganisms-13-01042-f004:**
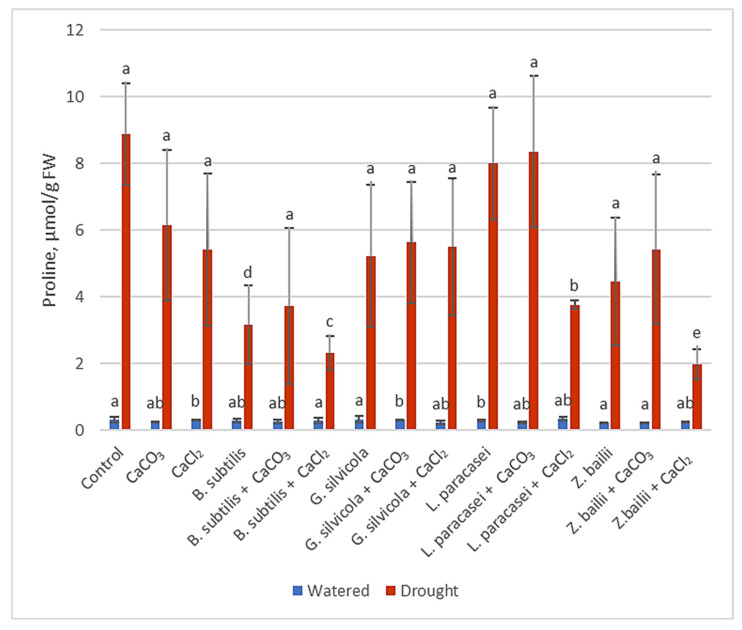
The effect of PGPMs and Ca salts application and prolonged drought stress on winter wheat proline accumulation. Error bars represent the standard deviation of the mean. Different lowercase letters in the same color columns indicate statistically significant differences (*p* < 0.05).

**Figure 5 microorganisms-13-01042-f005:**
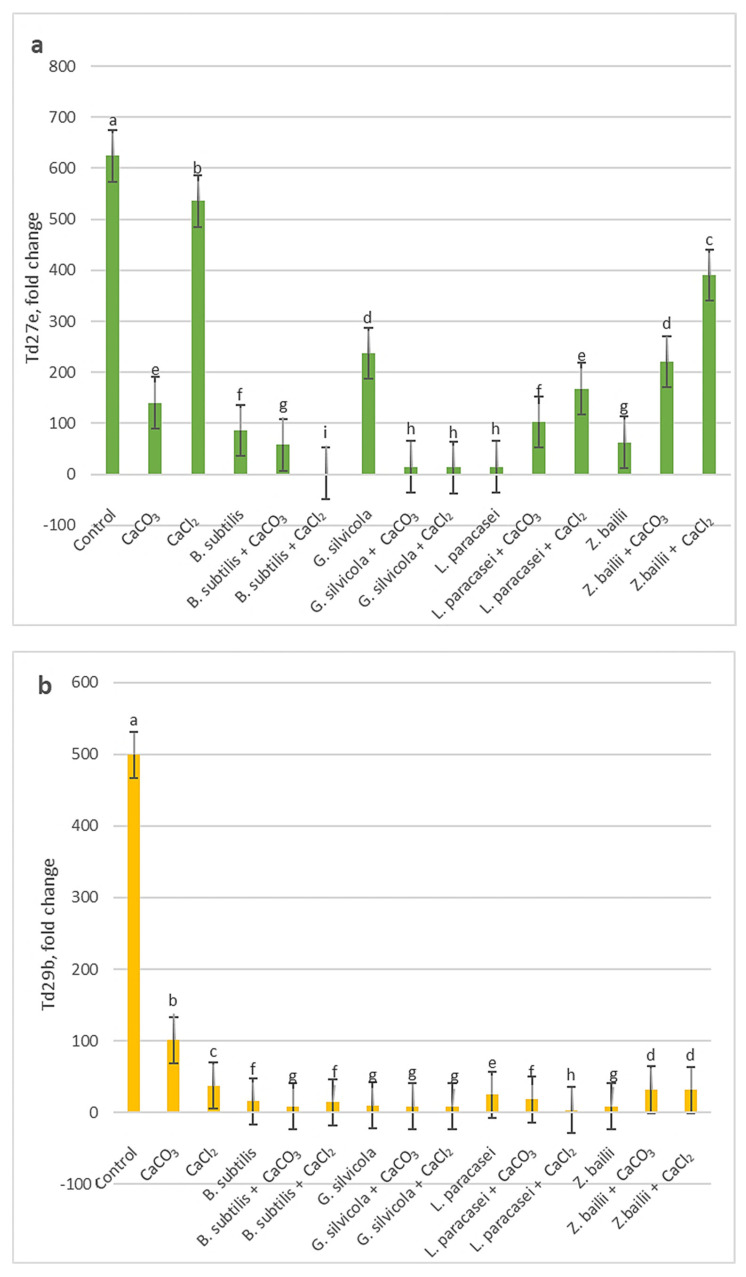
The effects of PGPMs and Ca salts application and prolonged drought stress on winter wheat lea genes ((**a**) *Td27e*; (**b**) *Td29b*; (**c**) *Td11*) expression level. Different lowercase letters in different gene fold changes indicate statistically significant differences (*p* < 0.05).

**Table 1 microorganisms-13-01042-t001:** Experimental design.

Treatment	CaCO_3_	CaCl_2_	Microorganisms
Control	−	−	−
CaCO_3_	+	−	−
CaCl_2_	−	+	−
*Bacillus subtilis*	−	−	+
*B. subtilis* + CaCO_3_	+	−	+
*B. subtilis* + CaCl_2_	−	+	+
*Geotrichum silvicola*	−	−	+
*G. silvicola* + CaCO_3_	+	−	+
*G. silvicola* + CaCl_2_	−	+	+
*Lactobacillus paracasei*	−	−	+
*L. paracasei* + CaCO_3_	+	−	+
*L. paracasei* + CaCl_2_	−	+	+
*Zygosaccharomyces bailii*	−	−	+
*Z. bailii* + CaCO_3_	+	−	+
*Z. bailii* + CaCl_2_	−	+	+

**Table 2 microorganisms-13-01042-t002:** The sequences of primers used in the work.

Gene	LEA Proteins Group	Primer Pairs	Primer Sequences (5′-3′)
*26S*		F/R	CCGGTTGTTATGCCAATAGCA/GCGGCGCAGCAGTTCT
*Td11*	2	F/R	AGGTGATCGATGACAACGGTG/ACCCTCGGTGTCCTTGTGG
*Td29b*	4	F/R	CGCACCCAGCTAGTAAGTTCG/CCCAGCCCAGTAATAACCCAT
*Td27e*	2	F/R	CAGCACTGAGCCGACGG/ACGTGGAACTAGAAGGCATTGAC

**Table 3 microorganisms-13-01042-t003:** Effect of PGPMs and Ca salts on morphometric parameters of winter wheat seedlings (per plant). Different letters in columns designate statistically significant differences at *p* < 0.05.

	Average Shoot Length, cm		Average Shoot Biomass, g	
	Watered	Drought	Watered	Drought
Control	24.2 d	15.95 f	0.288 e	0.039 f
CaCO_3_	26.95 b	17.05 e	0.3159 c	0.036 f
CaCl_2_	24.55 d	17.75 e	0.267 f	0.0335 f
*Bacillus subtilis*	27.95 a	20.65 b	0.333 b	0.1278 b
*B. subtilis* + CaCO_3_	27.55 a	21.65 a	0.345 a	0.1164 c
*B. subtilis* + CaCl_2_	27.85 a	19.73 c	0.3197 c	0.1218 b
*Geotrichum silvicola*	26.6 b	20.5 b	0.29 e	0.107 d
*G. silvicola* + CaCO_3_	26.9 b	19.56 c	0.297 d	0.109 d
*G. silvicola* + CaCl_2_	27.1 ab	18.45 d	0.316 c	0.112 c
*Lactobacillus paracasei*	25.3 c	18.55 d	0.269 f	0.068 e
*L. paracasei* + CaCO_3_	26.55 b	18.85 d	0.306 d	0.057 e
*L. paracasei* + CaCl_2_	25.8 c	20.45 b	0.253 f	0.134 a
*Zygosaccharomyces bailii*	25.1 c	18.65 d	0.2846 e	0.072 e
*Z. bailii* + CaCO_3_	26.45 b	16.4 f	0.309 d	0.035 f
*Z. bailii* + CaCl_2_	27.4 a	20.5 b	0.289 e	0.1454 a

## Data Availability

The data supporting the reported results can be found in the Nature Research Centre’s archive of scientific reports.

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
