# Peer review of "Physiological, Biochemical, and Genetic Reactions of Winter Wheat to Drought Under the Influence of Plant Growth Promoting Microorganisms and Calcium"

_microorganisms, 2025, doi:10.3390/microorganisms13051042_

Round 1
Reviewer 1 Report
Comments and Suggestions for Authors
Submission ID: microorganisms-3577688
Title of the manuscript: "Physiological, Biochemical, and Genetic Reactions of Winter Wheat to Drought under the Influence of Plant Growth Promoting Microorganisms and Calcium ".
Dear authors, thanks for the opportunity to review the manuscript. I really appreciate this well written and interesting review; it is a hot topic, and the authors did a lot of work. I suggest several points should be addressed by the authors as follows:
Specific comments
Point 1: L20-25, The authors should present the key data of the measured indicators, and p- values. This is important to provide concise results.
Point 2: L28-29, Please rearrange the keywords alphabetically, as well as define PGPM.
Point 3: L36-37, Not clear. Which studies??? What about their recommendations based on the results???
Point 4: L56, Why did the authors use the term biostimulants, why not bioinoculants??? Please clarify.
Point 5: L92-101, Before this paragraph, please add more details on the novelty of the study, you may refer to some lack in the previous study regarding some aspects.
Point 6: L203-204, The software or program is mandatory information.
Point 7: L206, Please make sure that P is italic in the whole manuscript.
Point 8: Table 3, Where are the standard deviation (SD) values according to your methodology???
Point 9: Figure 6, Move the treatments down a bit to show the negative error bars.
Point 10: I suggest citing Tables and Figures in the discussion section, to make the readers in contact with the results.
Point 11: L303, the authors should give a convincing explanation for proline increase due to the studied application under drought stress. It is thought that proline can act as a stress marker in addition to its role as an Osmo protectant. It means that plants may accumulate more proline due to a biotic stress or less proline as a result of reduced stress. Hence, the low proline accumulation in the treated plants indicated that these plants were less exposed to drought and did not require excessive proline accumulation to withstand drought stress. This paper will help you https://doi.org/10.3390/microorganisms12061123
Point 12: L352, Please make sure that the scientific names are italic in the whole manuscript.
Point 13: L350, Please delete this sentence “All treatments have a generally positive effect in drought conditions”. Try to give a clear recommendation from the practical side aspects.
Point 14: L359, The recommendations of future research are missing.
Respectfully, reviewer
Author Response
Response to Reviewer 1 Comments
Point 1: L20-25, The authors should present the key data of the measured indicators and p-values. This is important to provide concise results.
Response 1: Thank you for the comment. Since we have multiple treatments and many parameters, it’s hard to include more in the abstract, but following the recommendations, we did add some of the key data the p-value.
Point 2: L28-29, Please rearrange the keywords alphabetically, as well as define PGPM.
Response 2: Thank you for your comment. We edited the text according to your recommendations.
Point 3: L36-37, Not clear. Which studies??? What about their recommendations based on the results???
Response 3: Thank you for your valuable comment. During the manuscript’s edition, a sentence was accidentally removed. We edited the text, and now we hope you will find your answer.
Point 4: L56, Why did the authors use the term biostimulants, why not bioinoculants??? Please clarify.
Response 4: Thank you for the comment. From the results of our studies, we noticed that our treatments not only help the plants to respond to drought but also stimulate their growth, as you can also see from Table 3. That’s why were decided to use the term biostimulants over bioinoculants.
Point 5: L92-101, Before this paragraph, please add more details on the novelty of the study; you may refer to some lack in the previous study regarding some aspects.
Response 5: Thank you for the valuable comment. We added information that indicates the novelty of this work.
Point 6: L203-204, The software or program is mandatory information.
Response 6: Thank you for the comment. The name of the software was added.
Point 7: L206, Please make sure that P is italic in the whole manuscript.
Response 7: Thank you for your work and attention; it helped us to see what we missed.
Point 8: Table 3, Where are the standard deviation (SD) values according to your methodology???
Response 8: Thank you for your comment. In the graphics, it was easy to show standard deviation by using error bars, but as for Table 3 it was decided just to show the statistical significance by p-value letters to not make it more complicated with additional numbers of standard deviation.
Point 9: Figure 6, Move the treatments down a bit to show the negative error bars.
Response 9: Thank you for your attention to detail. We moved the names of treatments down for better visibility.
Point 10: I suggest citing Tables and Figures in the discussion section, to make the readers in contact with the results.
Response 10: Thank you for the valuable comment. We added the citations in discussion section.
Point 11: L303, the authors should give a convincing explanation for proline increase due to the studied application under drought stress. It is thought that proline can act as a stress marker in addition to its role as an Osmo protectant. It means that plants may accumulate more proline due to a biotic stress or less proline as a result of reduced stress. Hence, the low proline accumulation in the treated plants indicated that these plants were less exposed to drought and did not require excessive proline accumulation to withstand drought stress. This paper will help you https://doi.org/10.3390/microorganisms12061123
Response 11: Thank you for your work and help. We added a little more information concerning our results of proline test in the discussion section. We didn’t explain the proline multifunctional role in plants in more details here because we already did that in our earlier publication https://doi.org/10.3390/plants12061301
Point 12: L352, Please make sure that the scientific names are italic in the whole manuscript.
Response 12: Thank you for your attention. We rechecked them once again and fixed where needed.
Point 13: L350, Please delete this sentence “All treatments have a generally positive effect in drought conditions”. Try to give a clear recommendation from the practical side aspects.
Response 13: Thank you for your comment. We tried to reformulate it.
Point 14: L359, The recommendations of future research are missing.
Response 14: Thank you for your comment. It was added.

Reviewer 2 Report
Comments and Suggestions for Authors
Summary: The manuscript titled « Physiological, Biochemical, and Genetic Reactions of Winter Wheat to Drought under the Influence of Plant Growth Promoting Microorganisms and Calcium” investigated the influence of probiotic microorganisms in combination with calcium salts on the physiological and biochemical metabolic pathways of wheat exposed to drought stress and on the analysis of gene expression levels that contribute to wheat drought tolerance. The authors carried out the experiments and handled the data appropriately. However, the manuscript presents some issues in the quality of the preparation. See specific comments. Moreover, the English language must be revised thoroughly.
Specific comments:
Introduction: The Introduction correctly places the study in its context and clearly states its purpose.
Materials and methods:
-I suggest removing the figure 1. The same for Table 2.
- On what basis were the selected microorganisms chosen for this study? Have they been previously characterized for plant growth-promoting (PGP) traits? Please specify.
- What was the reasoning behind using both calcium carbonate and calcium chloride in this study, and how do these forms differ in their expected impact on drought stress mitigation in wheat? It would be helpful to provide further explanation on this aspect in M&M section.
-Line 116: witch culture media were used for strains growth? Please specify
-Line 144: It is recommended to define the acronyms upon their initial use in the formula.
-Lines 194-200: you could combine this part with the previous one (Real-Time Quantitative PCR) and the information in the table 2 could be added directly on the text.
Results and discussion:
-You should present the results in table 3 as mean ± deviation standard.
-The quality of figures should be ameliorated, and the statistical test should be mentioned in the captions.
-Lines 278-280: This part is more introductive.
Conclusion: Beyond a summary of the study, the authors should underline the importance of the findings obtained and future research directions.
Other comments:
-The English language should be revised for correctness and fluency.
-You should define the acronym “MDA” when you cite it for the first time.
-Adjust references list according to MDPI style (see references: 13-16 and 38)
-Line 352: Correct the italics in “B. subtilis”
Comments on the Quality of English Language-The English language should be revised for correctness and fluency.
Author Response
Response to Reviewer 2 Comments
Point 1. I suggest removing the figure 1. The same for Table 2.
Response 1: Thank you for your comment. We removed Figure 1, but as for Table 2, we find it the easiest way to present the primers of the genes that have been used in this study therefore, we decided to leave it.
Point 2. On what basis were the selected microorganisms chosen for this study? Have they been previously characterized for plant growth-promoting (PGP) traits? Please specify.
Response 2: Thank you for your questions. You will find the answer to your first question in the beginning of the discussion section. We previously worked with a commercial product that contains these or similar microorganisms, and they proved to leave biostimulating effect on plants under drought conditions. Especially, Bacillus subtilis is widely known for its PGP potential.
Point 3. What was the reasoning behind using both calcium carbonate and calcium chloride in this study, and how do these forms differ in their expected impact on drought stress mitigation in wheat? It would be helpful to provide further explanation on this aspect in M&M section.
Response 3: Thank you for your question. We previously worked just with CaCO3 and noticed that its presence affects differently when used in the field and in small pots in the laboratory. CaCl2 was chosen as a more easily absorbed form of Ca.
Point 4. Line 116: witch culture media were used for strains growth? Please specify
Response 4: Thank you for your mark here. We added some information about medias in the methodology section.
Point 5. Line 144: It is recommended to define the acronyms upon their initial use in the formula.
Response 5. Thank you for the comment, they are defined in the text right before the formula.
Point 6. Lines 194-200: you could combine this part with the previous one (Real-Time Quantitative PCR) and the information in the table 2 could be added directly on the text.
Response 6. Thank you for the comment, slight changes were done in that section.
Point 7. You should present the results in table 3 as mean ± deviation standard.
Response 7. Thank you for you comment. We answered concerning that to Reviewer 1 (see above).
Point 8. The quality of figures should be ameliorated, and the statistical test should be mentioned in the captions.
Response 8. Thank you for your comment. The figures were slightly corrected. The test is mentioned in the statistical analysis section.
Point 9. Lines 278-280: This part is more introductive.
Response 9. Thank you for your comment. We rechecked it.
Point 10. Conclusion: Beyond a summary of the study, the authors should underline the importance of the findings obtained and future research directions.
Response 8. Thank you for your valuable comment. The conclusion was edited.
Other comments:
-The English language should be revised for correctness and fluency.
-You should define the acronym “MDA” when you cite it for the first time. Thank you. Done
-Adjust references list according to MDPI style (see references: 13-16 and 38). Thank you for your work and attention. We changed them.
-Line 352: Correct the italics in “B. subtilis” Thank you. Done

Reviewer 3 Report
Comments and Suggestions for Authors
Please specify why cv. ‘Skagen’ was chosen.
Please report the soil composition.
Row 114 were the microorganisms used all together?
Put species names in italic.
Please see the attached file for other revisions and citation suggestions.

Author Response
Response to Reviewer 3 Comments
Please specify why cv. ‘Skagen’ was chosen.
Thank you for your comment. cv. ‘Skagen’ is widely used in Lithuania and has a fast growth in the laboratory conditions.
Please report the soil composition.
Soil was purchased from the local producer SuliFlor called SF1(a peat moss substrate (pH 5.5 – 6.5)).
Row 114 were the microorganisms used all together?
All of them were used separately in this study (however, we did study them in combination to each other).
Put species names in italic.
Thank you. We rechecked and fixed where needed.
Please see the attached file for other revisions and citation suggestions.
Thank you. We looked at it and corrected where needed.

Round 2
Reviewer 1 Report
Comments and Suggestions for Authors
This is the second time I have evaluated this manuscript. The authors addressed all my comments, and the manuscript has been noticeably improved. Many thanks for their contribution.
Reviewer 2 Report
Comments and Suggestions for Authors
The authors have addressed all the reviewers' comments/suggestions. I think the manuscript is now suitable for publication in the Microorganisms journal.